# Inverted device architecture for high efficiency single-layer organic light-emitting diodes with imbalanced charge transport

Xiao Tan [1], Dehai Dou [1], Lay-Lay Chua [2,3], Rui-Qi Png[2], Daniel G. Congrave [4], Hugo Bronstein [4,5], Martin Baumgarten[1], Yungui Li [1] ✉, Paul W. M. Blom [1] & Gert-Jan A. H. Wetzelaer [1] ✉

Many wide-gap organic semiconductors exhibit imbalanced electron and hole transport, therefore efficient organic light-emitting diodes require a multilayer architecture of electron- and hole-transport materials to confine charge recombination to the emissive layer. Here, we show that even for emitters with imbalanced charge transport, it is possible to obtain highly efficient single-layer organic light emitting diodes (OLEDs), without the need for additional charge-transport and blocking layers. For hole-dominated emitters, an inverted single-layer device architecture with ohmic bottom-electron and top-hole contacts moves the emission zone away from the metal top electrode, thereby more than doubling the optical outcoupling efficiency. Finally, a blue-emitting inverted single-layer OLED based on thermally activated delayed fluorescence is achieved, exhibiting a high external quantum efficiency of 19% with little roll-off at high brightness, demonstrating that balanced charge transport is not a prerequisite for highly efficient single-layer OLEDs.

Organic semiconductors exhibiting electroluminescence have been successfully applied in display technology and show promise for the next generation of solid-state lighting. The development of organic light-emitting diodes (OLEDs) started with early observations of electroluminescence in anthracene crystals[1]. A breakthrough was achieved by Tang and VanSlyke in 1987[2] when they demonstrated a double-layer OLED, which resulted in a significant drop in operating voltage. This multilayer strategy was widely developed over the next decades, where organic layers were inserted with different functionalities, such as charge injection, charge transport, and charge and exciton blocking[3,4]. As such, the supply of electrons and holes to the central emissive layer could be balanced, with the blocking layers ensuring that all charges and excitons recombine within the emissive layer, and thereby maximizing the internal quantum efficiency for photon generation. Furthermore, the emissive layer could be placed at an optically optimal position within the multilayer stack, resulting in maximum light extraction from the optical microcavity. The development of the

multilayer OLED architecture, along with the harvesting of triplet excitons with phosphorescent or thermally activated delayed fluorescence (TADF) emitters has led to OLEDs with high external quantum efficiencies (EQEs)[5,6], practically only limited by the light-outcoupling efficiency of typically 20-30%[7].

Despite the successful development of multilayer OLEDs, the multilayer architecture has some major drawbacks. First of all, many organic materials and deposition steps are required to fabricate such an OLED, which is costly and excludes the possibility of inkjet-printed OLEDs. Second, all layers and materials have to work in conjunction, to accomplish effective charge injection, charge transport, and charge and exciton confinement within the emissive layer, which makes the design and material selection cumbersome. Third, the many heterojunctions in the device lead to voltage losses and may be potential sources of device degradation. Fourth, understanding of the device physics is complicated, since the many layers and heterojunctions introduce unknown parameters, such as heights of the internal barriers

[1]Max Planck Institute for Polymer Research, Mainz, Germany. [2]Department of Physics, National University of Singapore, Singapore, Singapore. [3]National University of Singapore, Department of Chemistry, Singapore, Singapore. [4]Department of Chemistry, University of Cambridge, Cambridge, UK. [5]Cavendish Laboratory, University of Cambridge, Cambridge, UK. ✉e-mail: yungui.li@mpip-mainz.mpg.de; wetzelaer@mpip-mainz.mpg.de

and charge transport properties, making the device analysis and optimization a trial-and-error process.

Motivated by tackling these disadvantages, we have recently developed a highly efficient single-layer OLED based on a neat film of the TADF emitter 9,10-bis(4-(9H-carbazol-9-yl)−2,6-dimethylphenyl)−9,10-diboraanthracene (CzDBA)[8] situated between ohmic electron and the hole contacts, providing direct charge injection into the emitter[9]. This single-layer OLED exhibited a maximum EQE of 19% and low driving voltages. The high efficiency can be ascribed to the combination of ohmic charge injection, the high photoluminescence quantum yield of the emitter, and balanced electron and hole transport. As opposed to multilayer OLEDs, only the balance in charge transport of the emitter controls the position of the emission zone within the single-layer OLED and is therefore a major factor in the optical outcoupling efficiency[10].

Unfortunately, most organic semiconductors exhibit highly imbalanced charge transport with differences between the electron and hole transport up to several orders of magnitude. Recently, we found that charge transport is closely related to the energy levels of the organic semiconductor concerning a trap-free energy window spanning approximately 2.4 eV[11]. As opposed to narrow-gap organic semiconductors as used in efficient organic photovoltaics, mostly obeying this energy window, the wide-gap materials needed for OLEDs rarely fit within this energy window. An immediate consequence is that charge transport in OLED emitters is often imbalanced due to charge trapping, especially in blue emitters having a wide energy gap of close to 3 eV. From an energy-level perspective, most known TADF emitters have their ionization energy (IE) inside the trap-free window (< 6.0 eV), whereas the electron affinity (EA) is outside. As a result, charge transport in most current emitters is expected to be hole-dominated, with electron transport being limited by charge trapping. Using the known (hole-dominated) emitters in a single-layer OLED would therefore often result in a recombination zone close to the metallic cathode, which leads to direct quenching of excitons, as well as coupling of photons to surface-plasmon-polariton (SPP) modes[12]. Consequently, light extraction would be highly suppressed[13]. This would imply that for most TADF emitters a high efficiency in a single-layer architecture is not possible.

A potential solution to shift the recombination zone away from the metallic top electrode in a single-layer OLED is to invert the device structure. In this case, the faster holes are injected into the emitter from the top metallic electrode, while the slower electrons are injected from the bottom transparent electrode[14]. However, it is still unclear if this is a viable solution, or if imbalanced charge transport excludes efficient single-layer OLEDs fundamentally. For a single-layer OLED, efficient charge injection via ohmic contacts is a crucial prerequisite,

since blocking layers are absent[9]. For an inverted OLED, a low work function transparent bottom electrode is thus required[15]. Several attractive options for such bottom electrodes have been published, among which is a ZnO electrode with a thin polyethyleneimine layer to reduce the work function down to 3.1 eV[16]. Although such an injection structure has been applied successfully in combination with a high-EA emitter, it is unclear if the work function is low enough to achieve an ohmic electron contact with emitters with a low EA[17,18]. Another possibility to achieve electron injection in an inverted OLED is the use of n-type dopants, but the options for dopants with a sufficiently low IE are limited[19]. Recently, a family of self-compensated n-doped polymers was developed with effective work functions as low as 2.4 eV, based on solution-casting of air-stable materials followed by *in-device* activation on the n-doped state[20]. These n-doped polymers have been successfully applied in conventional OLED structures with electron injection from the top electrode. Recently, bottom electron-injecting blue inverted multilayer quantum-dot LEDs have been demonstrated with such n-doped polymers, in which charge confinement is achieved with additional blocking layers[21]. However, it is still unclear if these n-doped polymers can also be successfully applied in inverted OLEDs in the single-layer architecture without charge-transport or confinement layers.

Here, we demonstrate that highly efficient single-layer OLEDs can be realized, despite imbalanced charge transport of the emitter. For hole-dominated emitters, this is accomplished in an inverted OLED architecture, where efficient electron injection from the transparent bottom electrode is achieved with an n-doped polymer layer. By inverting the device structure, the charge-recombination zone is moved away from the metal top electrode, eliminating severe light loss due to the coupling of photons to SPP modes. Based on this concept, blue inverted single-layer OLEDs with an EQE of up to 19% are demonstrated, using an emitter with imbalanced transport and in the complete absence of high-triplet-energy host or blocking layers. In Fig. 1a, the device layout of our inverted single-layer OLED is shown. For electron injection, the bottom electrode is coated with the n-doped polymer TFB[20] (poly[[(9,9-di-n-octylfluorene-2,7-diyl)-alt-(1,4-phenylene-(4-sec-butylphenylimino)−1,4-phenylene)]) using spin coating. Although TFB is often associated with hole transport, the polymer can be n-doped *in-device* by oxalate electron transfer induced by hole sensitization in a self-compensated polymer to an ultralow effective work function of 2.4 eV. The work function of these self-compensated polymers is set by the electronic structure of the polymer semiconductor backbone together with the local coulomb (Madelung) potential effects of the ions, both counter-balancing and spectator[22,23]. For hole injection, we use a layer of the high-work function metal oxide $MoO_3$. To ensure ohmic charge injection, thin (3-

a)

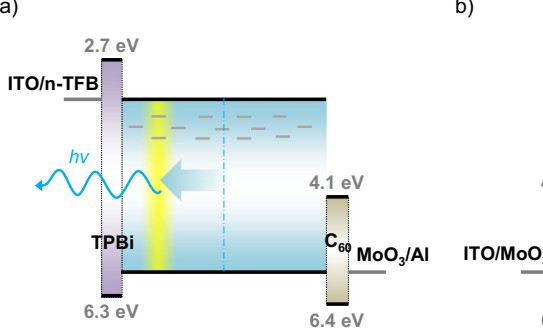

b)

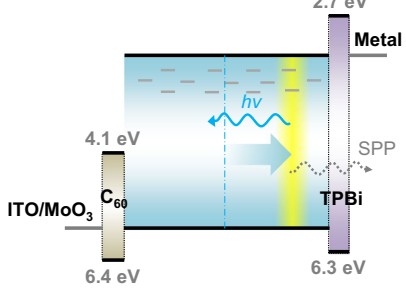

**Fig. 1 | Device design principle. a** Schematic energy band diagram of the inverted single-layer OLED. The emissive layer is sandwiched between an $MoO_3$/Al top anode and an ITO/TFB bottom cathode, using a thin $C_{60}$ and TPBi interlayer for the formation of an ohmic hole and electron contact, respectively. The diagrams schematically show the presence of electron traps, which slow down electron transport. As a result, the emission zone (yellow) shifts toward the electron-injecting contact. Photon losses to SPP modes are indicated in (**b**) for the conventional structure.

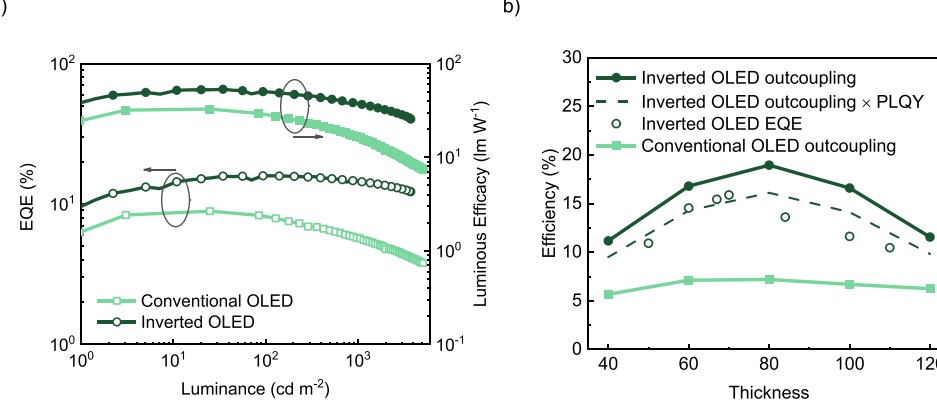

**Fig. 2 | Device performance of single-layer DMAC-BP OLEDs. a** EQE (open symbols) and power efficiency (closed symbols) as a function of luminance for conventional (light green) and inverted (dark green) OLEDs. **b** Emissive layer thickness-dependent outcoupling efficiency (closed symbols) simulated at an applied voltage of 2.7 V, the dashed line represents the product of the inverted OLED outcoupling efficiency and the PLQY, indicating the maximum predicted EQE, with the experimental EQE displayed as open symbols.

4 nm) tunneling interlayers of TPBi and $C_{60}$ are being used at the respective electrodes. These interlayers decouple the electrode and the organic semiconductor electrostatically, restoring the Fermi-level alignment[24]. It should be noted that while still multiple layers are being used, these are part of the charge-injection structure. These thin layers are transparent for charges[24] and are too thin to prevent energy transfer of excitons[25]. Charge-transport and blocking layers are absent, which in conventional multilayer OLEDs have to be tuned to the emitter in terms of energy levels and triplet energy, which is not the case here. Here, the OLED functionally has a single-layer structure, since charges are directly injected into the emitter and thus the position of the emission zone is controlled by the balance in charge-transport properties of the emitter. This balance should not be confused with the "charge-balance factor" in multilayer OLEDs[26], which is a measure of the recombination yield of electrons and holes. We note that in a single-layer OLED, the high density of majority carriers near the ohmic contacts ensures that minority carriers cannot exit the device without recombining, resulting in negligible leakage of charges[27].

## Results

To test the newly designed inverted device structure, we first selected the yellow TADF emitter CzDBA[8] for the emissive layer. CzDBA has been successfully applied in a conventional single-layer OLED and has fairly balanced charge transport[28]. As such, it is excellently suited to assess the proper functioning of the charge-injection structure. The measured current density and luminance as a function of voltage and EQE are shown in Supplementary Fig. 2. The EQE reaches a maximum of 15.5%, which is a decent value, but somewhat lower compared to the conventional OLED structure, which reaches an EQE of 19%[9]. The lower EQE can be explained by the electron transport being slightly superior to the hole transport in CzDBA, resulting in a recombination zone closer to the metallic top electrode in an inverted OLED, which slightly reduces optical outcoupling. To put the sensitivity to charge injection into perspective, a single-layer CzDBA OLED with a conventional non-ohmic LiF/Al cathode reaches an EQE of only 0.3%[9]. Therefore, the high EQE for the inverted device indicates that electron injection is efficient.

With the successful fabrication of an inverted single-layer CzDBA OLED, the next step is to apply the inverted device structure to an emitter with a lower EA and imbalanced charge transport. To this end, we selected the green TADF emitter bis[4-(9,9-dimethyl-9,10-dihydroacridine)phenyl]methanone (DMAC-BP) with an IE of 5.8 eV and an EA of 3.1 eV[29,30]. This emitter has been successfully used in a non-doped multilayer OLED, reaching a maximum EQE of 18.9%[29]. However, in the

same study, a single-layer DMAC-BP OLED achieved an EQE of only 0.06%, which is expected to be due to a combination of imbalanced charge transport and suboptimal charge injection. Since the charge-transport characteristics of DMAC-BP are unknown, we fabricated hole-only and electron-only devices with ohmic contacts. The measured current density-voltage characteristics are shown in Supplementary Fig. 3a. The hole current is orders of magnitude higher than the electron current, indicating imbalanced charge transport. The charge-transport properties were quantified by fitting the hole and electron currents with numerical drift-diffusion simulations[31]. The electron and hole mobility amounted to $1 \times 10^{-7}$ cm$^2$ V$^{-1}$ s$^{-1}$ and $4 \times 10^{-7}$ cm$^2$ V$^{-1}$ s$^{-1}$, respectively. However, while hole transport is trap-free, the electron transport is severely reduced due to the presence of electron traps with a density of $2 \times 10^{23}$ m$^{-3}$, which can be seen from the steep voltage-dependence of the current, before reaching the trap-filled limit. The presence of electron traps, rather than the mobility of free electrons, results in an electron current that is orders of magnitude lower than the hole current. This is unlike the situation in emitters with balanced transport, such as CzDBA, where the electron and hole current are almost equal due to the near absence of charge traps. As a result of the highly imbalanced charge transport due to severe electron trapping in DMAC-BP, bimolecular recombination in a single-layer DMAC-BP OLED occurs close to the cathode interface, as shown by the drift-diffusion simulations of the recombination profile in Supplementary Fig. 3b. Therefore, it is expected that an inverted OLED architecture could be of benefit to the performance in case of DMAC-BP as the emitter.

As a next step, DMAC-BP is applied in both conventional OLED (Fig. 1b) and inverted OLED (Fig. 1a) structures with an emissive layer thickness of 70 nm. The inverted OLED shows a maximum EQE of 15.9%, compared to 8.9% for the conventional OLED configuration, as shown in Fig. 2a. The higher efficiency of the inverted device is a direct consequence of the superior hole transport, enhancing the optical outcoupling by shifting the recombination zone away from the metallic top electrode. Furthermore, the inverted OLED shows very little efficiency roll-off at high brightness. The higher efficiency roll-off for the conventional device structure is attributed to the emission zone moving closer to the metallic cathode at higher driving voltages, as shown by the simulations in Supplementary Fig. 3b. This effect is absent in the inverted OLED, as the bottom cathode is nonmetallic and therefore does not lead to coupling to SPP modes. It is noted that the electroluminescence spectrum of the inverted OLED is broadened by 7 nm (Supplementary Fig. 2d), which we consider to be due to the different positions of the emission zone concerning the optical microcavity. It is worth mentioning that the efficiency of the inverted

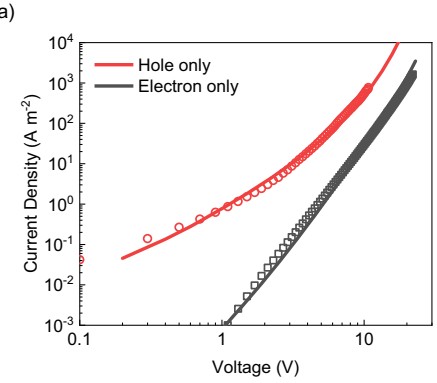

a)

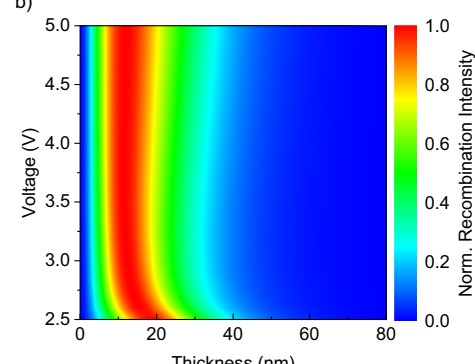

b)

**Fig. 3 | Charge transport in 2tCz2CzBN and simulated recombination profile.** **a** Current density–voltage characteristics of 2tCz2CzBN electron- (97 nm) and hole-only (108 nm) devices (symbols). Solid lines are fits with a numerical drift-diffusion model (red for hole-only and black for electron-only devices). **b** The voltage-dependent recombination profile normalized to the total rate for a 2tCz2CzBN OLED with an 80 nm emissive layer.

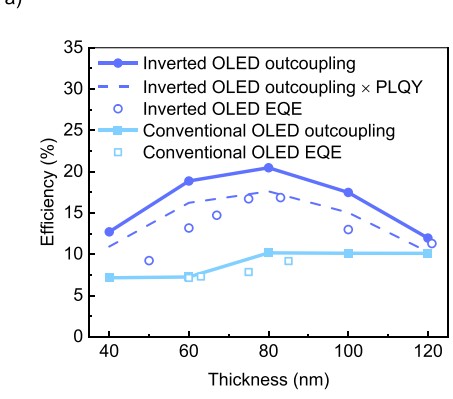

a)

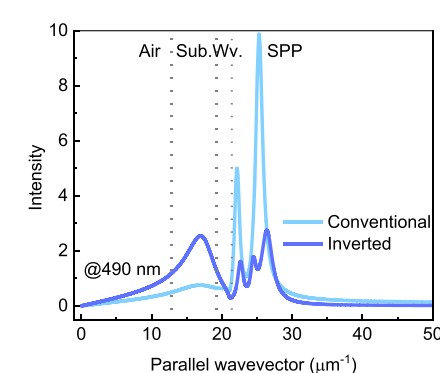

b)

**Fig. 4 | Optical outcoupling efficiency simulation for 2tCz2CzBN single-layer OLEDs.** **a** Emissive layer thickness-dependent outcoupling efficiency (closed symbols), the dashed line represents the product of the inverted OLED outcoupling efficiency and the PLQY, indicating the maximum predicted EQE, with the experimental EQE displayed as open symbols. **b** Power-dissipation plot for a conventional (light blue) and an inverted single-layer OLED (blue). In the inverted OLED, the power dissipated into SPP modes is substantially reduced.

single-layer OLEDs is not far off the efficiency of published multilayer DMAC-BP OLEDS[29].

The improvement in EQE by inverting the device structure is also rationalized by optical outcoupling simulations. Using a previously developed procedure, the optical outcoupling efficiency into air can be simulated based on the calculated recombination profile[10]. The recombination profile is simulated based on the experimental charge-transport parameters, as outlined above. The optical-outcoupling model includes the measured optical constants of all layers and the orientation of emitting dipoles is measured by the angular dependence of photoluminescence, as presented in Supplementary Fig. 4. As is displayed in Fig. 2b, the maximum outcoupling efficiency for an inverted DMAC-BP device equals 19%. The measured EQE of 15.9% therefore indicates a high internal quantum efficiency, when considering the photoluminescence quantum yield (PLQY) of neat DMAC-BP of 85%[29].

Furthermore, the thickness dependence of the outcoupling efficiency is shown in Fig. 2b. For single-layer OLEDs, changing the layer thickness modifies the optical cavity, while the recombination profile for any thickness can be simulated with the same set of charge transport parameters[10,32]. The experimental thickness-dependent EQE follows the same trend as the simulated optical outcoupling efficiency and is quantitatively well described when further considering the PLQY of the DMAC-BP neat film. The agreement in the thickness dependence confirms that the simulated recombination profile, based on the experimental transport parameters, is accurate. Qualitatively, one can understand the thickness dependence considering that a maximum in outcoupling is achieved when emission takes place at a specific distance from the metal, which is about 70 nm for a green emitter[33]. Since the optimal layer thickness is close to this value, the recombination zone must be close to the transparent cathode. The outcoupling efficiency for the conventional device is calculated to be only 7.2%, slightly lower than the experimental EQE. A possible explanation for this discrepancy could be that the outcoupling model is highly sensitive to the exact position of the emission zone when it is in close vicinity to the metal interface[13]. In that case, a minor inaccuracy in the exact recombination profile may lead to a substantial difference in the simulated outcoupling efficiency for the conventional device architecture, in which the recombination zone is very close to the cathode.

The results of the green inverted DMAC-BP OLED demonstrate that efficient single-layer OLEDs are feasible even for emitters with highly imbalanced charge transport. Taking into account the energy-level considerations of the trap-free window, imbalanced charge transport appears to be a major problem especially for blue emitters, having a wide energy gap of close to 3 eV. Therefore, as a next step, we investigate the feasibility of an inverted blue single-layer OLED, despite imbalanced charge transport. Here, we chose 2tCz2CzBN, a blue TADF emitter with an IE of 6.0 eV, showing promising performance (EQE$_{max}$ of 21.6%) in an undoped multilayer OLED[34]. The reported EA of 3.1 eV is expected to be accessible for electron injection from n-TFB. It should

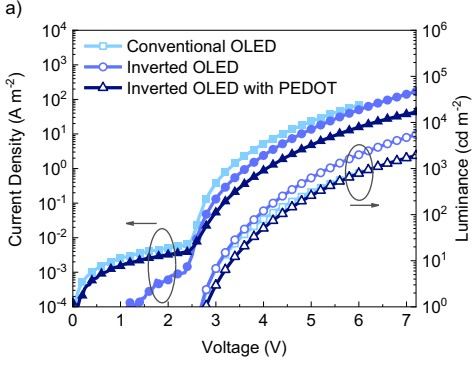
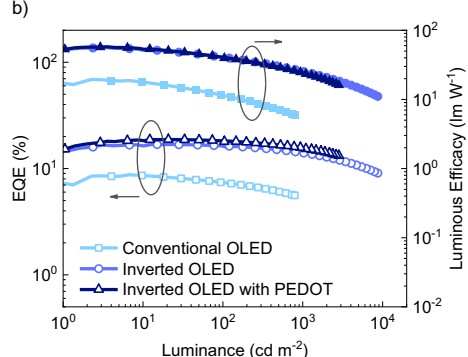

**Fig. 5 | Device performance of single-layer 2tCz2CzBN conventional and inverted OLEDs. a** Current density–voltage (closed symbols) and luminance–voltage (open symbols) characteristics of a 2tCz2CzBN single-layer conventional OLED (85 nm) in light blue, inverted OLED (83 nm) in blue and inverted OLED with an additional PEDOT: PSS layer (83 nm) in dark blue with optimal thickness for each device structure. **b** EQE (open symbols) and power efficiency (closed symbols) as a function of luminance.

**Table 1 | Device performance of OLEDs based on different TADF emitters**

| Device | $V_{on}$[b)] [V] | $EQE_{max}$[c)] [%] | $EQE$[d)] [%] | $CE_{max}$[e)] [cd A$^{-1}$] | $PE_{max}$[f)] [lm W$^{-1}$] | $\lambda_{EL}$[g)] [nm] |
|---|---|---|---|---|---|---|
| CzDBA inverted | 2.6 | 15.5 | 14.0 | 53.2 | 50.2 | 560 |
| DMAC-BP conventional | 2.6 | 8.9 | 5.7 | 27.6 | 32.7 | 511 |
| DMAC-BP inverted | 2.6 | 15.9 | 14.5 | 52.8 | 53.2 | 518 |
| 2tCz2CzBN conventional | 2.7 | 8.7 | – | 17.9 | 18.7 | 490 |
| 2tCz2CzBN inverted | 2.7 | 16.9 | 14.1 | 52.3 | 57.0 | 490 |
| 2tCz2CzBN inverted[a)] | 2.8 | 18.7 | 15.5 | 58.1 | 57.7 | 490 |

[a)]Inverted OLED based on 2tCz2CzBN with an additional 40 nm PEDOT:PSS layer on top of the ITO cathode, [b)] Turn-on voltage ($V_{on}$) at 1 cd m$^{-2}$, [c)] External quantum efficiency (EQE) at maximum and [d)] at 1000 cd m$^{-2}$, [e)] Current efficiency (CE) and [f)] Power efficiency at their maximum, [g)] Electroluminescence peak wavelength.

be noted that direct evaluation of the electron-injection properties of n-TFB is not possible in an electron-only device, as hole sensitization is required for n-doping[20]. However, as shown in Supplementary Fig. 5, the built-in voltages of the inverted and conventional OLEDs are equal, implying that the n-TFB contact is close to ohmic. Using the same procedure as outlined previously for DMAC-BP, the charge transport properties (Fig. 3a) of 2tCz2CzBN were obtained. Also in this case, hole transport is superior to electron transport, as observed from the substantial difference in electron and hole current. The extracted electron and hole mobility amounted to $2 \times 10^{-9}$ cm$^2$ V$^{-1}$ s$^{-1}$ and $6 \times 10^{-9}$ cm$^2$ V$^{-1}$ s$^{-1}$, respectively. While hole transport is trap free, electron transport is limited by traps with a density of $4 \times 10^{23}$ m$^{-3}$, giving rise to highly imbalanced transport. Similar to the case of DMAC-BP, the imbalance in charge transport is not so much caused by a difference in mobility, but by the presence of a considerable electron-trap density. These charge-transport parameters are subsequently used to simulate the voltage-dependent recombination profile numerically in a double-carrier device, as shown in Fig. 3b. The recombination zone is, as expected, close to the cathode. Furthermore, the recombination zone shifts closer to the cathode with increasing driving voltage, while slightly broadening due to the spreading of the charge concentration.

Based on the simulated recombination profile, the optical-outcoupling efficiency is calculated for different emissive layer thicknesses, as shown in Fig. 4a. For the inverted OLED, a maximum outcoupling efficiency to air of 20% is determined for the optimal emissive layer thickness range of 60–80 nm. For the conventional OLED, an outcoupling efficiency of up to only 7% is simulated. The outcoupling efficiency is slightly higher than for the DMAC-BP inverted OLED, as a result of a higher fraction of horizontally oriented emitters, with a measured anisotropy factor of 0.254 in terms of vertical dipoles (Supplementary Fig. 4b).

Figure 4b shows the optical power dissipated inside the inverted and conventional device structure at the peak wavelength, while the power dissipation for all wavelengths is displayed in Supplementary Fig. 6. These results demonstrate that the fraction of photons coupled into non-radiative SPP modes is drastically reduced in the inverted structure, in which the emission zone is no longer in close vicinity to the metal top electrode. This is the main reason for the increased efficiency of the inverted single-layer OLEDs, exhibiting a much higher fraction of generated photons coupled into air.

Based on the optical simulations, inverted and conventional single-layer blue OLEDs based on 2tCz2CzBN were fabricated and the device performance is shown in Fig. 5. The inverted single-layer blue OLED shows an impressive EQE of 16.9%, which can even be increased to 18.7% when using an additional poly(3,4-ethylenedioxythiophene) polystyrene sulfonate (PEDOT:PSS) layer to further tune the device optics underneath the TFB layer. The PEDOT:PSS layer does not influence the built-in voltage (Supplementary Fig. 5) and thus does not compromise the electron injection. However, the device current is slightly lower, possibly due to the added series resistance of the layer. The high EQE confirms the maintained electron injection, as even a small injection barrier would drastically compromise the efficiency[9]. This EQE approaches the maximum outcoupling efficiency and points to a very high internal quantum efficiency, considering the measured PLQY for a 2tCz2CzBN film of 86%. Furthermore, the efficiency roll-off at high brightness is very minor, showing that efficient blue single-layer OLEDs at high luminance are feasible. At 1000 cd m$^{-2}$, the corresponding EQE of 15.5% for the inverted OLED is even higher than the reported value (10.8%[34]) for a multilayer OLED with the same emitter. Conversely, the conventional single-layer OLED shows a maximum EQE of 8.7%, which is expected based on the outcoupling simulations. The electroluminescence spectra are shown in Supplementary Fig. 5c with

maxima at 490 nm. The device-performance parameters of all OLEDs in this study are summarized in Table 1.

Figure 4a shows that the experimental EQE as a function of emissive layer thickness follows the trend of the simulated optical outcoupling efficiency. Compared to inverted devices, both the measured EQE and simulated outcoupling efficiency of conventional devices are less sensitive to thickness. The main reason is that the recombination zone is always close to the metallic cathode, implying that changing the layer thickness will not induce major changes in outcoupling efficiency. By contrast, in the inverted OLEDs, changing the layer thickness directly shifts the position of the recombination zone concerning the metal electrode, to which outcoupling is very sensitive. Overall, the agreement between the simulated outcoupling efficiency and the experimental EQE as a function of layer thickness confirms the calculated position of the recombination zone.

As we have demonstrated that efficient single-layer OLEDs are feasible, even with imbalanced charge transport of the emitter, a question that may arise is whether imbalanced charge transport would affect operational stability. Fundamentally, single-layer OLEDs have shown increased lifetimes compared to multilayer OLEDs, due to a broadened recombination zone. In Supplementary Fig. 9a, it is demonstrated that the conventional OLED based on DMAC-BP has a lifetime that is a factor of ~2.5 higher than a previously reported multilayer OLED based on a non-doped DMAC-BP emissive layer[29]. As the electron-injection layer for the inverted structure is not yet optimized for stability, the lifetime of these devices is reduced, as discussed in Supplementary Note 1. Nevertheless, it is apparent from the conventional single-layer OLEDs that imbalanced transport and the resulting recombination zone still lead to stability advantages compared to multilayer devices, demonstrating that the single-layer concept is a promising route towards more stable OLEDs.

In summary, we have demonstrated a strategy to design highly efficient single-layer OLEDs despite imbalanced charge transport of the emitter. For the most common case of a hole-dominated emitter, this is achieved with a newly designed inverted device structure. Due to the superior hole transport, the emission zone is far from the metallic hole-injecting electrode, thereby enhancing light extraction by suppressing photon coupling into SPP modes. A blue single-layer OLED with an EQE of 18.7% with a small roll-off at high brightness was demonstrated, on par with multilayer OLEDs. This demonstrates that balanced transport is not a prerequisite to realize highly efficient single-layer blue OLEDs, extending the single-layer OLED concept to a wide range of TADF emitters of all colors, without the need for high-triplet-energy blocking and host layers.

## Methods
### Materials
2tCz2CzBN was synthesized according to procedures in the literature and purified by vacuum sublimation. Details are given in the Supplementary Information. CzDBA, DMAC-BP, TPBi, and $C_{60}$ were purchased in sublimed grade from Ossila BV. Chemicals for synthesis were purchased from common suppliers (Sigma-Aldrich etc.) and were used as received.

### Device fabrication
Pre-patterned indium-tin-oxide(ITO)-covered glass substrates were cleaned with the detergent solution and ultrasonicated in acetone and subsequently in isopropyl alcohol for 5 min. The substrates were then heated at 140 °C for 10 min and subsequently treated with UV-ozone for 20 min. For some devices, PEDOT:PSS (Heraeus Al 4083) layers of 45 nm were spin-coated on the ITO substrates, which were annealed at 140 °C for 10 min subsequently. For inverted OLEDs, a solution of n-TFB was prepared using a TFE(2,2,2-trifluoroethanol):OFP(2,2,3,3,4,4,5,5-octa-fluoro-1-pentanol) solvent mixture in a volume ratio of 3:1, which was

spin-cast in a nitrogen-filled glovebox to form a film with a thickness of 14 nm, on top of either ITO or PEDOT:PSS. Organic tunneling interlayers of 4 nm ($C_{60}$ and TPBi) were thermally evaporated under high vacuum. Layers of $MoO_3$ (10 nm), Ba (5 nm), and Al (100 nm) were thermally evaporated under high vacuum. For electron-only devices, glass substrates were used, with a thermally evaporated Al (30 nm) bottom electrode.

### Measurements
Electrical characterization was carried out under nitrogen atmosphere with a Keithley 2400 source meter and light output was recorded with a Si photodiode with NIST-traceable calibration, with a detector area (1 cm²) larger than the emitting area of the OLED (0.16 cm²). The photodiode was placed close to (but not in contact with) the OLED to capture all photons emitted in a forward hemisphere. To avoid any light detection emitted from the substrate edges, the edges were masked by the sample holder and the substrate size (3 × 3 cm²) was considerably larger than the photodetector area. The EQE, luminance, and power efficiency were calculated from the measured photocurrent, the device current, and the electroluminescence spectrum. Electroluminescence spectra were obtained with a USB4000-UV–VIS-ES spectrometer.

### Simulations
The voltage-dependent recombination profiles were obtained based on drift-diffusion simulations using the extended Gaussian disorder model (EGDM). The parameters are summarized in Supplementary Table 2. The refractive index and dipole orientation factor for the DMAC-BP and 2tCz2CzBN neat film were experimentally determined by ellipsometry and angular-dependent photoluminescence. For all the other materials, published complex refractive indices determined by experimental measurements were used.

## Data availability
Source data willl become available at https://doi.org/10.6084/m9.figshare.25664445

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

## Acknowledgements

The authors thank Prof. Peter K.H. Ho for stimulating discussions. D.G.C. acknowledges the Herchel Smith fund for an early career fellowship. X.T. acknowledges the China Scholarship Council (No.202008320380). The authors thank the technical support from Frank Keller, Christian Bauer, Michelle Beuchel, and Sirma Koynova.

## Author contributions

G.-J.A.H.W. proposed the project. G.-J.A.H.W. and X.T. designed the experiments. X.T. carried out the experiments, simulations and wrote the manuscript. Y.L. performed the optical simulations. D.D. and D.G.C. synthesized the compound 2tCz2CzBN, H.B. and M.B. supervised the synthetic work, L.C. and R.P. synthesized the compound n-doped TFB, G.-J.A.H.W. and P.W.M.B. supervised the project.

## Funding

## Competing interests

The authors declare no competing interests.
