## [Peer Review File · Nature Communications]

Inverted device architecture for high efficiency single-layer organic light-emitting diodes with imbalanced charge transportEditorial Note: This manuscript has been previously reviewed at another journal that is not operating a transparent peer review scheme. This document only contains reviewer comments and rebuttal letters for versions considered at *Nature Communications*.

REVIEWER COMMENTS

Reviewer #1 (Remarks to the Author):

The author has largely incorporated the reviewer's suggestions and provided thoughtful responses. However, I believe there are three points where clarity is still lacking:

1. The author's response primarily focuses on the enhanced stability of single-layer devices compared to traditional multi-layer devices. Yet, in this study, the inverted device exhibited significantly lower stability than conventional devices (see Figure S8), and was even less stable than previously reported multilayer devices. The reason behind this discrepancy remains unclear. The authors should offer some explanations of how the structure of the device impacts its lifetime.
2. The structure of the electron-only device in this study did not include TFB or a PEDOT:PSS/TFB combination as the injection layer, diverging from the design typically used in inverted OLEDs. This difference could lead to inaccuracies in the measured electron injection efficiency and transport rate. The authors are encouraged to address and amend this discrepancy.
3. It is unclear whether the PEDOT:PSS layer in the inverted device serves solely to enhance light outcoupling efficiency, or if it also influences the device's lifetime and the balance of carriers. The authors need to clarify the multifaceted role of the PEDOT:PSS layer.

Reviewer #2 (Remarks to the Author):

In the revised paper, the authors have given reasonable response for most of question, but the next question should be discussed.

- 1, The functions of TPBI and C60 are used in inverted device, so it is not reasonable for the concept "single-layer blue organic light-emitting diodes".
- 2, About the lifetime of inverted device, the authors should give more data analysis compared to conventional device.

Reviewer #3 (Remarks to the Author):

The authors have answered all the questions raised by reviewers.

The revision is satisfactory, and it can be accepted for publication.

Reviewer #4 (Remarks to the Author):

Here, I am re-evaluating the present manuscript NCOMMS-24-05332-T, which is a revision through transfer based on 4 referee reports. I am evaluating the response to my original comments (= referee 4). Overall, the authors have addressed all of my comments in detail. Thank you very much. I have two comments, where the first one relates to an original comment I made and the second one is now. Assuming that those are addressed in an additional minor revision, I am happy to recommend this manuscript for publication.

1. RE: Comment 3, Referee 4: As outlined, the actual devices discussed in the manuscript are of various color, but not purely blue from a display usage point of view. The title suggests differently. In the reply, the authors even state that they don't see any reason, why their concept should not work for other colors. Therefore, I strongly suggest to remove the term 'blue' from the title. It will even give an opportunity to be seen as general concept, rather than a 'blue'-only progress.

2. While re-visiting the manuscript after the revision of the authors, I stumbled across the classification of balanced and imbalanced emission layers in these OLEDs. I am sure that I understand the difference in bottom and inverted structure leading to a different position where the recombination takes place - and this is due to imbalanced transport - I fully agree. Still, very well established in the OLED community is the so-called charge balance factor in the external quantum efficiency equation. $EQE = [\text{charge balance factor}] * [\text{PLQY}] * [\text{singlet/triplet emitter} \rightarrow 0.25 \text{ or } 1] * [\text{outcoupling}]$. In this definition, the charge balance is rather a measure for charges that leak through the device without recombination (e.g., if an electron travels through the complete OLED, it will be counted as contribution to the current but does not contribute to the exciton formation). Here, I see the two usages of 'balanced/imbalanced' to mean different things. In the work of the authors, their imbalanced transport to the sight of recombination leads to a specific position of the recombination zone, but it does not affect the charge balance factor in the EQE formula. This is obviously not the case, as the authors report very high EQE.  Therefore, expecting that this double use of similar terms might confuse the broader readership, I would kindly ask

the authors to add a paragraph to properly work out the different meaning of 'balance/imbalance' I pointed out before.

Response Letter

We thank all reviewers for their careful review. Please find the full reviewer reports, along with our point-by-point response (in green) below.

Reviewer #1 (Remarks to the Author):

The author has largely incorporated the reviewer's suggestions and provided thoughtful responses. However, I believe there are three points where clarity is still lacking:

1. The author's response primarily focuses on the enhanced stability of single-layer devices compared to traditional multi-layer devices. Yet, in this study, the inverted device exhibited significantly lower stability than conventional devices (see Figure S8), and was even less stable than previously reported multilayer devices. The reason behind this discrepancy remains unclear. The authors should offer some explanations of how the structure of the device impacts its lifetime.

The reviewer is right that we highlighted the enhanced stability of single-layer devices compared to multilayer devices. We believe that it is an important result that imbalanced transport does not compromise the stability advantages of single-layer OLEDs. However, we do agree with the reviewer that more explanations are required for the reduced stability of the inverted OLED. With the added discussion along with Fig. S9, we were able to identify that the electron-injection layer (n-TFB) is limiting the stability. This can be inferred from the stable voltage until half life, which in these hole-dominated OLEDs implies that the degradation is on the electron side. As the only difference in the device structure is the electron-injection layer, we conclude that electron injection is reduced during degradation. The voltage rise near end of life may indicate failure of n-TFB to extract holes (possibly de-doping). As the hole contact is stable, as confirmed by stressing hole-only devices (Fig. S9c), the late voltage rise is not due to reduced hole injection or transport.

We do however note that lifetime is not the primary focus of this study and we believe it can be greatly improved in future work.

2. The structure of the electron-only device in this study did not include TFB or a PEDOT:PSS/TFB combination as the injection layer, diverging from the design typically used in inverted OLEDs. This difference could lead to inaccuracies in the measured electron injection efficiency and transport rate. The authors are encouraged to address and amend this discrepancy.

As correctly observed by the reviewer, n-TFB was not used in the electron-only devices. The reason is that n-TFB switches on via a hole-sensitization mechanism (<https://doi.org/10.1038/s41586-019-1575-7>), implying that holes are required to achieve electron injection. This is unfortunately not possible in an electron-only device. However, we can determine the effective work function from the built-in voltage of the OLEDs, as now shown in Fig S5 a). As the built-in voltage is the same for the inverted and conventional OLEDs, this implies that the difference between the hole and electron quasi-Fermi levels at the contacts is equal. This in turn confirms that n-TFB is a (near-) ohmic electron contact. A second observation to confirm that the contact is ohmic comes from the EQE: even a small injection barrier drastically reduces the EQE. The high observed EQE is only possible for a good electron contact.

3. It is unclear whether the PEDOT:PSS layer in the inverted device serves solely to enhance light outcoupling efficiency, or if it also influences the device's lifetime and the balance of carriers. The authors need to clarify the multifaceted role of the PEDOT:PSS layer.

We have included the device lifetime data of inverted OLED with the PEDOT:PSS layer in Fig S9, which shows similar lifetime behavior. From the V_{bi} analysis in Figure S5 a), it is observed that the PEDOT:PSS layer does not influence the built-in voltage and thus does not compromise the electron injection. However, the device current is slightly lower, possibly due to the added series resistance of the layer (a resistance in series does not impact the balance of carriers, it only reduces the voltage drop across the emissive layer). Please note that these are hole-dominated devices (because of the transport imbalance), so the electron injection itself does not have a significant impact on the magnitude of the device current.

Reviewer #2 (Remarks to the Author):

In the revised paper, the authors have given reasonable response for most of question, but the next question should be discussed.

1, The functions of TPBi and C60 are used in inverted device, so it is not reasonable for the concept "single-layer blue organic light-emitting diodes".

We thank the reviewer for the comments. As we have mentioned in the section discussing the device concept, the TPBi and C60 interlayers form part of the injection structure. In the revised version, we have now elaborated on this, stating that these layers are completely transparent for charges (<https://doi.org/10.1038/s41563-018-0022-8>) and are too thin to prevent resonant energy transfer of excitons (<https://doi.org/10.1063/1.464927>). As a result, these layers do not add any charge-blocking or

exciton-blocking functionality (C_{60} itself even rapidly quenches excitons and is a very strong electron acceptor). As a result, these layers only have the purpose of charge injection, while the charge transport and recombination is completely controlled by the emissive layer. This is in stark contrast to a multilayer OLEDs, where charge-transport and blocking layers determine the position of the recombination zone. The purpose of the paper is to demonstrate how the (im)balance in charge transport of the emitter controls the outcoupling, which is applicable to single-layer devices, not to multilayer devices.

2, About the lifetime of inverted device, the authors should give more data analysis compared to conventional device.

As suggested by the reviewer, we have added additional analysis. First, we have added the voltage behavior upon stressing under constant current. The constant voltage until half life reveals that the main cause of degradation of these hole-dominated OLEDs is on the electron side. As the only difference between the conventional and inverted devices is the electron-injection layer (n-TFB), the electron injection reduces upon stressing. Since the device current is anyway predominantly carried by holes, a reduction in electron injection does not affect the current significantly, meaning that under constant current stressing the voltage remains constant. The voltage rise near end of life may indicate failure of n-TFB to extract holes (possibly de-doping). In a second experiment, stressing hole-only devices, the hole contact is observed to be stable, showing that the late voltage rise in OLEDs is not due to reduced hole injection or transport. Stressing of the hole-only device additionally reveals that the hole contact is stable in both the top and bottom configuration.

Reviewer #3 (Remarks to the Author):

The authors have answered all the questions raised by reviewers.

The revision is satisfactory, and it can be accepted for publication.

We thank the reviewer for the positive comments.

Reviewer #4 (Remarks to the Author):

Here, I am re-evaluating the present manuscript NCOMMS-24-05332-T, which is a revision through transfer based on 4 referee reports. I am evaluating the response to my original comments (= referee 4).

Overall, the authors have addressed all of my comments in detail. Thank you very much. I have two comments, where the first one relates to an original comment I made and the second one is now. Assuming that those are addressed in an additional minor revision, I am happy to recommend this manuscript for publication.

1. RE: Comment 3, Referee 4: As outlined, the actual devices discussed in the manuscript are of various color, but not purely blue from a display usage point of view. The title suggests differently. In the reply, the authors even state that they don't see any reason, why their concept should not work for other colors. Therefore, I strongly suggest to remove the term 'blue' from the title. It will even give an opportunity to be seen as general concept, rather than a 'blue'-only progress.

We thank the reviewer for the nice suggestion, indeed the concept is more general, rather than limited to a certain color range. Therefore, the title has been adjusted as recommended.

2. While re-visiting the manuscript after the revision of the authors, I stumbled across the classification of balanced and imbalanced emission layers in these OLEDs. I am sure that I understand the difference in bottom and inverted structure leading to a different position where the recombination takes place - and this is due to imbalanced transport - I fully agree. Still, very well established in the OLED community is the so-called charge balance factor in the external quantum efficiency equation. $EQE = [\text{charge balance factor}] * [\text{PLQY}] * [\text{singlet/triplet emitter} \rightarrow 0.25 \text{ or } 1] * [\text{outcoupling}]$. In this definition, the charge balance is rather a measure for charges that leak through the device without recombination (e.g., if an electron travels through the complete OLED, it will be counted as contribution to the current but does not contribute to the exciton formation). Here, I see the two usages of 'balanced/imbalance' to mean different things. In the work of the authors, their imbalanced transport to the sight of recombination leads to a specific position of the recombination zone, but it does not affect the charge balance factor in the EQE formula. This is obviously not the case, as the authors report very high EQE.  Therefore, expecting that this double use of similar terms might confuse the broader readership, I would kindly ask the authors to add a paragraph to properly work out the different meaning of 'balance/imbalance' I pointed out before.

We thank the reviewer for the detailed clarification of the different definitions of the terms 'balance/imbalance'. We agree that the terminology may lead to confusion and we have therefore included a brief discussion in the manuscript (in the 'device concept' section) to clarify the differences. We have also included the reason why the recombination efficiency is so high in these single-layer OLEDs.

REVIEWERS' COMMENTS

Reviewer #1 (Remarks to the Author):

The revised manuscript was well-improved according to the reviewer's suggestions, and becomes acceptable in this journal.

Reviewer #2 (Remarks to the Author):

The authors have given reasonable response, it is suggested to be accepted by NC

Reviewer #4 (Remarks to the Author):

In this revision, the authors have address the two remaining comments of my 2nd round review, so that now all concerns are eliminated. I can happily recommend this improved version for publication.